# Feasibility of Early Assessment for Psychological Distress: HRV-Based Evaluation Using IR-UWB Radar

**DOI:** 10.3390/s24196210

**Published:** 2024-09-25

**Authors:** Yuna Lee, Kounseok Lee, Sarfaraz Ahmed, Sung Ho Cho

**Affiliations:** 1Department of Electronic Engineering, Hanyang University, Seoul 04763, Republic of Korea; yuna0722@hanyang.ac.kr (Y.L.); sarfaraz687@hanyang.ac.kr (S.A.); 2Department of Psychiatry, College of Medicine, Hanyang University, Seoul 04763, Republic of Korea; dual@hanyang.ac.kr

**Keywords:** radar, impulse radio ultra-wideband radar, IR-UWB radar, mental distress, psychological distress, heart rate variability, HRV

## Abstract

Mental distress-induced imbalances in autonomic nervous system activities adversely affect the electrical stability of the cardiac system, with heart rate variability (HRV) identified as a related indicator. Traditional HRV measurements use electrocardiography (ECG), but impulse radio ultra-wideband (IR-UWB) radar has shown potential in HRV measurement, although it is rarely applied to psychological studies. This study aimed to assess early high levels of mental distress using HRV indices obtained using radar through modified signal processing tailored to reduce phase noise and improve positional accuracy. We conducted 120 evaluations on 15 office workers from a software startup, with each 5 min evaluation using both radar and ECG. Visual analog scale (VAS) scores were collected to assess mental distress, with evaluations scoring 7.5 or higher classified as high-mental distress group, while the remainder formed the control group. Evaluations indicating high levels of mental distress showed significantly lower HRV compared to the control group, with radar-derived indices correlating strongly with ECG results. The radar-based analysis demonstrated a significant ability to differentiate high mental distress, supported by receiver operating characteristic (ROC) analysis. These findings suggest that IR-UWB radar could be a supportive tool for distinguishing high levels of mental stress, offering clinicians complementary diagnostic insights.

## 1. Introduction

Heart rate variability (HRV) is the variation in time between consecutive heartbeats, used as a non-invasive marker of autonomic nervous system activity [1]. HRV, influenced by autonomic nervous system activity, is linked to cardiovascular diseases (e.g., heart failure, sudden death), as well as stress and psychiatric conditions [2]. Various analytical methods, including linear and frequency domain analyses, have been utilized to observe HRV. Indices like the mean RR interval (mean RR) and the standard deviation of NN intervals (SDNN) have proven useful for psychiatric HRV analysis [3]. Psychiatric studies on HRV are typically linked to mental distress, encompassing emotional pain like sadness, worry, and anger [4]. Several research studies have found a link between the HRV indices and mental distress [5,6]. Previous studies have found associations between HRV and depressive or anxiety disorders [7]. Studies have also investigated cases where anxiety disorders are linked to decreased HRV [8]. Some studies have observed significant differences in the SDNN and high-frequency (HF) power spectral density of HRV in patients with various mental disorders [9]. Chronic stress has been reported to excessively activate the sympathetic nervous system, leading to negative changes in HRV [10]. Additionally, studies have reported that HRV is also linked to psychiatric conditions such as depression and anxiety [11,12]. HRV measurement is primarily conducted using electrocardiography (ECG), a widely used tool for assessing the heart’s electrical activity through electrodes placed on the skin [13]. However, ECG’s reliance on contact electrodes limits its use for long-term monitoring or in patients who cannot tolerate electrodes, such as those with severe burns [14]. Alternative HRV measurement techniques include photoplethysmography, used in smartwatches, and ballistocardiography, which measures the body’s response as the heart pumps blood [15]. Recent research has introduced radar sensors as a non-contact HRV measurement tool, overcoming the limitations of skin contact by providing a completely non-invasive method for assessing HRV parameters and physiological conditions [16,17].

Researchers have explored various methods to assess mental stress using radar technology. One study classified different mental states—such as normal, fatigue, stress, and sleep—using the mental arithmetic task (MAT) for stress assessment [18]. Mental distress detection using bioradar has been used to identify stress level by solving a simple mathematical problem [19]. A study has assessed major depressive disorder by analyzing HRV through 60 mental tasks, using PPG, ECG, and radar sensors to measure biomarkers [20]. One of the commonly used techniques involves the use of Doppler radar to measure HRV as a marker for stress [21]. An ultra-wideband radar is utilized to monitor respiration patterns, which change under stress [22]. Millimeter-wave radar systems are employed to capture multiple physiological signals simultaneously, including heart rate and stress-related changes [23].

Radar-based HRV measurement offers significant potential for real-world healthcare applications, including remote patient monitoring, stress assessment, athletic performance analysis, occupational health, and sleep quality evaluation. For instance, Li et al. (2019) used UWB radar to monitor HRV during simulated driving tasks, demonstrating its effectiveness in detecting driver drowsiness [24]. The results indicated that radar-based HRV measurements could reliably indicate fatigue and alertness levels in drivers. Additionally, Chen et al. (2020) investigated the use of millimeter-wave radar for assessing HRV and identifying depressive symptoms in patients [25]. The study reported promising results, highlighting radar-based HRV measurements’ potential in depression screening and monitoring. Doppler radar has also proven effective in monitoring driver fatigue by assessing physiological parameters like heart rate and respiration [26]. Ultra-wideband radar systems have been utilized to monitor stress and anxiety levels by tracking physiological signals such as HRV and respiration patterns [27]. Radar technology has been used to detect and monitor depression by analyzing HRV and other physiological signals [28]. Radar-based healthcare applications are transforming the way conditions such as fatigue, stress, depression, and anxiety are monitored and managed. The non-invasive nature and continuous monitoring capabilities of radar systems provide valuable data, enhancing early detection, intervention, and overall patient care [29,30].

Previous studies have mainly used radar technology to measure HRV through methods like the MAT for evaluating mental distress. However, none have evaluated mental distress through VAS scores along with HRV statistical parameters for employees who experience stress due to office workload. Our study focuses on full-time office workers at a software startup, exploring radar-based HRV measurement for long-term data collection and psychiatric research. This non-invasive method will offer clinicians additional information for more accurate diagnostics. To ensure the accuracy of HRV measurements, we compared radar-derived HRV indices with those from a reference ECG sensor and conducted measurements while participants were seated. Radar technology’s continuous monitoring capabilities make it suitable for various environments, including office settings. Our goal is to provide a tool for ongoing stress monitoring near workstations, identifying high-risk individuals. Through this study, we successfully measured HRV with radar and conducted preliminary validation, demonstrating that high mental distress groups can be distinguished using radar-based HRV measurements and VAS scores to assess stress levels in the workplace.

The objective of our paper is to confirm the feasibility of radar-based technology for measuring psychological mental distress. This novel approach assesses high levels of mental distress through HRV measurement using radar technology and VAS scores. Our contributions are threefold: First, we evaluated high-level mental distress caused by office workload using VAS scores and HRV statistical parameters. Second, we developed a customized radar HRV algorithm that effectively identifies HRV statistical parameters. Finally, and more importantly, we confirmed that radar-based measurement of HRV indices such as mean RR intervals and SDNN is effective in identifying high levels of mental distress.

Our research offers significant benefits for patients needing long-term monitoring where conventional ECGs are inconvenient or impractical. Radar-based psychological distress assessment provides clinicians with an additional tool to complement existing diagnostic methods. It delivers reliable data on autonomic nervous system activity, aiding in diagnosing mental distress when traditional data is scarce or difficult to obtain. This technology can help clinicians tailor stress management and mental health interventions, reducing burnout and improving mental health outcomes. Scalable and easy to implement, radar-based assessment can enhance accessibility to mental health evaluations, especially in remote or underserved areas, supporting telemedicine where traditional resources are limited.

This article is structured as follows: Section 2 delves into the materials and methods employed in our study, covering the study’s overall design, participant details, experimental setup, the ground truth used for validation, and the algorithm utilized for calculating HRV. Section 3 presents the experimental results. In Section 4, we discuss the implications of these findings. Finally, the paper concludes with Section 5, which provides a summary of our conclusions.

## 2. Materials and Methods

In this paper, we developed a comprehensive process to assess psychological distress, as illustrated in Figure 1. During the initial evaluation, we collected three types of data: radar data, ECG data, and VAS scores for mental distress. ECG and radar datasets were utilized to validate the radar’s performance by comparing HRV indices derived from both the radar and ECG measurements. Upon successful validation of the radar-based HRV’s accuracy, we proceeded to evaluate high levels of mental distress. Evaluations were categorized into two distinct levels based on their VAS scores. Those with the highest VAS scores (VAS ≥ 7.5) were classified as experiencing high mental distress. To explore the relationship between stress levels and radar-based HRV indices, we conducted a series of statistical analyses, including ANOVA and ROC analysis, focusing particularly on the differences between stress groups and radar-based HRV metrics.

### 2.1. Subjects

The study included individuals who were employed full-time at a software development company based in Seoul, Korea. The company is a software development startup company, primarily focused on developing software products or services. All participants in the study were office workers engaged in predominantly sedentary work, and all of them were full-time daytime employees. The participants were 23–44 years old with an average age of 30.9 years. Throughout the experiment, 120 evaluations were performed on 15 participants. Demographic information is provided in Table 1. The participants were healthy individuals without any pre-existing illnesses. They received explicit instructions to refrain from using tobacco products, alcohol, and caffeine for at least 8 h before the trial. The experiment took place within the past three months.

### 2.2. Experimental Setup

The experimental setup of the study is illustrated in Figure 2. The experiment was conducted in a quiet environment, minimizing external interference that could potentially impact the study outcomes. A total of 120 experiments were conducted on 15 participants. Each participant participated in the experiment a total of eight times: four times upon arriving at work and four times upon leaving work. This occurred twice daily over four weekdays for each individual. At the beginning of the experiment, participants were asked to answer a VAS questionnaire regarding their current mental distress level. After answering the questionnaire, the participant’s HRV was measured for 5 min using ECG and radar simultaneously. Consistent with previous research, the radar was fixed at an angle that allowed observation of the carotid artery from the right chest position of a seated subject [31]. The average distance between the radar transceiver pair and the patient’s chest was approximately 0.3 m. In this context, the “radar transceiver pair” refers to a single integrated radar unit that seamlessly performs both transmission and reception of signals. This indicates that the radar system, responsible for both transmitting signals and receiving their reflections from the patient’s chest, was positioned at this distance. Radar signals were received simultaneously with the ECG device used as a reference, utilizing USB serial communication. Signal collection was facilitated through USB serial communication using MATLAB R2022b (Math Works Inc., Natick, MA, USA).

For HRV measurement using radar, we used Novelda’s X4M02 module, an IR-UWB radar module (Novelda AS, Oslo, Norway) shown in Figure 2. The X4M02 module operates within a frequency range of 6.5–8.5 GHz, with a sampling frequency of 23.328 GHz, providing high-resolution measurements of approximately 1.5 cm. The module incorporates a single-input single output antenna configuration utilizing a pair of transceivers. To achieve a time resolution of 4 ms, the module was set to measure 250 frames per second (FPS). The maximum measurement range of the radar was set at 1.5 m to prevent additional noise from interfering with the signal. The radar’s start range was set to 0.4 m to utilize the reference phase of the signal, where the signal travels directly from the transmitter to the receiver without reflecting off the subject. While the module offers sub-millimeter accuracy, potential noise from multi-path reflections and environmental factors was minimized by the hardware manufacturer. Additionally, the X4M02 is equipped with advanced signal processing capabilities to further filter out environmental noise and interference.

### 2.3. Ground Truth

The aim of this study is to estimate levels of mental distress using the HRV indices obtained from the IR-UWB radar. To ensure the validity of our findings, we also validate the accuracy of the HRV indices measured with the IR-UWB radar compared to a conventional device. We establish two references, namely Reference 1 and Reference 2, which serve as benchmarks for our research.
Reference 1: ECG (for validating the accuracy of HRV measurements)Reference 2: VAS (for evaluating the participants’ mental distress)

Reference 1 served to validate the accuracy of HRV measurements obtained from the IR-UWB radar. To accomplish this, we employed a reliable HRV measurement method, namely an ECG. Specifically, we used the PSL-iECG2 module from Physio Lab (Busan, Republic of Korea). The PSL-iECG2 module followed the traditional approach of measuring ECGs using three electrodes. The ECG signals were collected via USB serial communication at a frame rate of 250 FPS (with a time resolution of 4 ms). Both radar and ECG signals were collected simultaneously with timestamps, eliminating the need for a separate synchronization process. No additional signal processing steps were applied to the collected ECG signal, except for the manufacturer’s own unspecified processing. Peak detection for the ECG signal was performed using the same algorithm described for the radar signal in Section 2.5. We conducted a comparison between the HRV indices derived from the collected ECG signals and the corresponding indices obtained from the IR-UWB radar. The HRV indices were the mean of RR intervals (Mean RR), the standard deviation of NN intervals (SDNN), the root mean square of successive difference of RR intervals (RMSSD), and the ratio between the absolute power of the low frequency and the high frequency bands (LF/HF ratio). The LF frequency region was 0.05 to 0.15 Hz, and the HF region was 0.15 to 0.40 Hz. The units of Mean RR, SDNN, and RMSSD are milliseconds (ms), while the LF/HF ratio is a dimensionless ratio.

Reference 2 was used in the evaluation of the participants’ mental distress. For this purpose, we selected the widely used VAS as it is a simple and reliable method for assessing stress levels [32,33]. The VAS involves a questionnaire in which participants rate their levels of conditions such as pain, mood, and worry by selecting a point on a line between two opposite extremes. In most cases, the distance from the leftmost starting point is used to calculate the scores. Participants’ VAS answers were requested for each trial, as shown in Figure 3. Specifically, participants were requested to rate their level of mental distress immediately prior to the HRV measurement, with a response resolution set at 0.5. The range of scores varied depending on the specific experiment. In this study, the experiments with response scores above the third quartile (7.5) on the VAS scale were defined as the “high-mental distress” group, as shown in Figure 3. The remaining experiments were classified as a control group and further divided into three sub-control groups according to the VAS score.

### 2.4. Radar Signal Processing and Algorithm

To derive HRV from IR-UWB radar, calculating RR intervals is essential. This was achieved through a signal processing technique, as depicted in Figure 4, which outlines the steps of the procedure. The commercial IR-UWB radar transmits radio waves received by the transceiver and converts them into a digital signal through an analogue-to-digital converter. The signal is then reflected by objects and returned. The acquired signal, referred to as “Received Raw Data” in Figure 4, is represented as a matrix with two axes: fast time and slow time. In this context, fast time is an expression in the time domain and carries the physical significance of time of arrival (ToA), which denotes the time it takes for radar pulses to be transmitted and received. Because (ToA) is directly proportional to distance, it is reasonable to consider the fast-time axis as the distance axis [34]. In contrast, slow time represents time in the conventional sense. In this experiment, we employed a full slow-time window of 5 min to calculate the RR intervals, which were subsequently used to derive the predetermined HRV indices. 

The received radar data may include signal distortion due to fast-time phase noise [35]. However, in most cases, when adequate radar parameter values are ensured, the impact of phase noise can be disregarded. However, in this study, the IR-UWB radar was employed in an extremely demanding environment with a frame rate of 250 FPS. The parameter values employed for noise reduction, including the number of signal smoothing iterations and the pulse-to-pulse interval, inevitably degraded compared to the typical scenario where the FPS range is 10 to 40. As a result, phase compensation and band pass filtering stages, as shown in Figure 4′s basic signal processing of the IR-UWB radar and phase noise compensation, were applied to the fast-time signal to counteract distortion [36]. Phase compensation involves selecting a reference phase that is presumed to be free of distortion. We then calculated the angle difference between this reference phase and another signal matrix to correct for phase distortion. In this study, the phase compensation used involved a straightforward calculation of phase differences for correction, which did not introduce any novel techniques.

Even after phase compensation, the radar signal matrix still exhibited interference from respiration, leading to unwanted peaks associated with respiratory activity. To mitigate this, we applied a series of filters. First, a smoothing filter with a cutoff frequency of 5 Hz and a filter order of 30 was used. Following this, we implemented a high-pass filter with cutoff frequencies of 40 and 60 Hz and a filter order of 1000. Together, these filters effectively functioned as a bandpass filter, successfully removing the respiratory component and unwanted noise. In Figure 5, the graphs titled “Phase Compensation” and “Bandpass Filtering” display intermediate signals of a 30 min plot of one fast-time point. These graphs demonstrate a significant enhancement in the clarity of the heartbeat signal after applying phase compensation and a series of filters. This process effectively reduces interference, making the heartbeat signal more distinct and prominent.

After basic signal processing, including phase compensation, it becomes possible to derive the relative sedentary movement value of the subject from the signal matrix [37]. To determine the presence of sedentary movement, we applied a fixed threshold and assessed it on a per-second basis. The thresholds for movement detection were determined using an enhanced constant false alarm rate algorithm, which utilized 5 min movement sequences [38]. Consequently, the total duration of sedentary movement was calculated for each experiment during the 5 min measurement period. To extract the slow-time “vital signal” containing heartbeat information, it was necessary to exclude the signals recorded during sedentary movement. When sedentary movements occurred, the radar signals were inevitably distorted due to their sensitivity to physical displacement. Consequently, it was essential to eliminate the periods affected by these movements. We recorded the timestamps of the sedentary movements and subsequently removed the peak intervals corresponding to these timestamps, as elaborated in Section 2.5. To maintain consistency, the same time periods were also excised from the ECG signals of the reference device.

When extracting the vital signal, it is crucial to identify the optimal fast-time point that encompasses the most relevant information. Typically, the fast-time point that exhibits the largest variance from the slow-time signal is chosen. However, at the fast-time point with the maximum slow-time variance, the respiratory component tends to dominate the slow-time signal. This is due to the chest displacement caused by respiration being generally more significant than that caused by the heartbeat. Consequently, the harmonic or inter-modulation frequency components associated with respiration can introduce distortions in HRV measurements [39,40]. Considering our experimental requirements of capturing detailed heart information, a different method for fast-time point selection was employed exclusively for HRV extraction. In this approach, the frequency information of the slow-time signal at each fast-time point was investigated. An adult heartbeat typically falls within the frequency range of 40/60–130/60 Hz (range HR) according to our testing. Hence, the spectral density of each fast-time point was examined to identify the most suitable fast-time point. We selected the fast-time point where the signal ratio of range HR to noise signal was the largest. By extracting the slow-time signal from the selected fast-time sample (MP), we obtained a vital signal presumed to contain the HRV information.

### 2.5. Heart Rate Variability (HRV)

To find the HRV in the vital signal, it is necessary to identify the peaks corresponding to each heartbeat. Because the vital signal is presumed to contain the heartbeat component, the peaks representing the chest/neck displacement caused by the heartbeat can be extracted from this one-dimensional signal. Each peak interval will exhibit a strong correlation with the RR interval obtained from the ECG. Hence, we considered the peak intervals obtained using radar as the RR intervals. 

To calculate the RR intervals in the vital signal acquired from radar, it is necessary to detect a series of peaks within the selected vital signal. Simple algorithms for peak detection in time-series data have been used in previous studies [41]. Instead, we adopted a new peak detection algorithm, which is a modified and simplified version of one previously used in clutter detection for people counting [29]. We do not detect peaks sequentially; instead, we detect them based on the amplitude of each peak, as outlined in Equation (1). The equation iterates continuously with increasing n, the iteration counter.
x_n_ = arg max (*v*[*k*])
y_n_ = *v*[x_n_](1)
*v*[x_n_ − R : x_n_ + R] = 0

In Equation (1), *v*[*k*] represents the vital signal extracted in Section 2.4, which is continuously updated in a loop. The variable x_n_ is the index of a local peak candidate, and y_n_ is the corresponding amplitude candidate of that local peak. R denotes the resolution of the peak detection. Equation (1) iterates until the entire update *v*[*k*] reaches entirely zero or the number of detected peaks exceeds a predefined limit. In this experiment, the peak detection limit was chosen based on the typical heart rate range of adults. Among the selected candidates x_1_…x_N_, we only selected the indices that satisfied in Equation (2).
y_n_ > max (*v*_0_[x_n_ − R : x_n_ + R])(2)

In Equation (2), *v*_0_ represents the original vital signal, which remains unchanged during the iteration. By applying in Equation (2), we could identify the true peaks that represented the actual local maxima. Once the selection of the real peak indices that satisfied the given conditions was complete, we sorted the real peak indices in chronological order. In Figure 5’s “Peak Detection” graph, a series of detected peaks is displayed, highlighting the effectiveness of the peak detection algorithm in identifying heartbeat events. By calculating the differences between the timestamps of consecutive peaks, an array of peak intervals could be derived for computing the HRV indices, as shown in Figure 6′s example. Figure 6 illustrates a 2 min example sequence of RR intervals obtained from both the radar and ECG signals, along with their respective power spectral density, which represents the frequency domain of HRV. The peak intervals derived from radar closely resemble the RR intervals obtained from the ECG, as depicted. Additionally, the frequency domain analysis (absolute FFT) of the RR sequences reveals similar trends. We used an FFT size of 2^18^ and applied a Hamming window of the same length as the RR interval array before performing the FFT. We did not apply interpolation to the omitted periods due to distortion caused by sedentary movement. We assumed the RR intervals to be continuous. 

However, to eliminate artifacts caused by sedentary movements, the peak intervals corresponding to the timestamps of these movements, as determined in Section 2.4, were excluded from the HRV indices calculation. Finally, the time-series peak intervals, after removing these artifacts, were utilized for the final calculation of the HRV indices. To ensure consistency, the same peak detection method and the elimination of sedentary movement-affected peak intervals were also applied to the ECG signals, which served as our reference device. 

### 2.6. Statistical Analysis

To assess the accuracy of the HRV measurement, the study compared four indices (mean RR, SDNN, RMSSD, and LF/HF ratio) calculated from 5 min RR interval data obtained from both radar and ECG. A total of 120 evaluations were analyzed; however, 18 invalid evaluations were excluded from the statistical analysis. Evaluations were considered invalid if the RMSSD range measured by the IR-UWB radar exceeded 100 or if the SDNN measured by the IR-UWB radar fell outside the range of SDNN obtained from the ECG (which ranged from 8.3 to 98.4 ms in this study) [42]. In addition to the HRV measurement, a VAS questionnaire for mental distress was completed for each experiment using ECG and IR-UWB radar. The 120 experiments were divided into four groups based on the VAS response results for a group comparison. Experiments with a response score above the third quartile (7.5) of the VAS scale were classified as the “high-mental distress” group, while the remaining experiments were categorized into three control groups (control 1, control 2, and control 3). Pearson’s correlation coefficient was also calculated to examine the correlation between VAS scores and HRV indices. For statistical analysis, MATLAB R2022 (Math Works Inc., Natick, MA, USA) and R 4.2.2 were used, and the R packages used were “readxl”, “Performance Analytics”, “pROC”, and “extrafont”.

## 3. Results

Before assessing mental distress, the first step was to verify the accuracy of the HRV measurements obtained by the radar. To assess the accuracy of HRV measured by radar, a correlation analysis was performed on 102 evaluations, excluding 18 outliers based on the previously described criteria. Figure 7 illustrates the comparison of HRV indices measured by radar and ECG. Each row in Figure 7 represents a specific HRV index: mean RR, SDNN, RMSSD, and LF/HF ratio. The graphs display a total of 102 experimental results, with each point representing a 5 min measurement. Note that we excluded RR intervals contaminated by sedentary movements for both radar and ECG, as detailed in Section 2.4 and Section 2.5. Across the 102 evaluations, the average duration of sedentary movement per evaluation was 7.24 s, with a standard deviation of 11.66 s.

The correlations between the HRV indices acquired from ECG (x-axis) and IR-UWB radar (y-axis) are shown in the first column of Figure 7 via scatter plots. The top and bottom two straight lines denote the 95 percent confidence interval (CI), whereas the middle straight line indicates the linear regression. The second column shows Bland–Altman plots, with the x-axis representing the average of the indices obtained from ECG and IR-UWB radar, and the y-axis representing the difference between them. The solid middle line represents the average offset from the ECG measurement of the HRV index measured by the IR-UWB radar, while the dotted line represents the 95 percent CI line. The correlation analysis revealed Pearson’s correlation coefficient values for the HRV indicators as follows: Mean RR (0.9997), SDNN (0.8962), RMSSD (0.6067), and LF/HF ratio (0.8234). The p-values for all four indices were less than 0.005, indicating a significant correlation between the measurements obtained from the ECG and the IR-UWB radar sensor. Figure 7 shows scatter and Bland–Altman plots for the HRV-based assessment of mental distress.

We also conducted an interclass correlation coefficient (ICC) analysis to compare radar-based and ECG-based heart rate variability (HRV) metrics, using a two-way random effects model to assess absolute agreement between the two methods. The results indicated near-perfect agreement for Mean RR, with an ICC of 0.9999 (95% CI: 0.9998–0.9999), excellent agreement for SDNN, with an ICC of 0.9342 (95% CI: 0.8713–0.9624), good to moderate agreement for RMSSD, with an ICC of 0.7027 (95% CI: 0.4554–0.8247), and strong agreement for the LF/HF ratio, with an ICC of 0.8307 (95% CI: 0.6088–0.9118).

### 3.1. HRV Indices and VAS Score

A total of 102 trials were performed, including all groups but excluding the previously mentioned outliers. The average VAS score was 4.691, with values ranging from a minimum of 0.0 to a maximum of 9.0, and quartiles at First Quartile (1st Qu) = 3.0, Median = 5.0, and Third Quartile (3rd Qu) = 6.0. The total control groups underwent 90 experiments whereas the high-mental distress (VAS 7.5) group underwent a total of 12 experiments. Table 2 provides data on HRV indices values for both ECG and IR-UWB radar measurements across different VAS score ranges. The groups are categorized as follows: Control 1 includes evaluations with a VAS score ranging from 0 to less than 2.5, with a sample size of 21; Control 2 encompasses those with a VAS score from 2.5 to less than 5.0, with a sample size of 20; and Control 3 covers evaluations with a VAS score from 5.0 to less than 7.5, with a sample size of 49. The combined group labeled as Control (all) includes evaluations with a VAS score ranging from 0 to less than 7.5, with a total sample size of 90. The high-mental distress group consists of evaluations with a VAS score ranging from 7.5 to less than 10, with a sample size of 12.

Figure 8 represents box plots depicting the HRV indices of ECG radar for the following two groups: Control (all) and high-mental distress group. The *p*-values for mean RR measured by radar and ECG were 0.0022 and 0.0020, respectively. The *p*-values for SDNN measured by radar and ECG were 0. 0127 and 0.0050, respectively. Similarly, the *p*-values for RMSSD measured by radar and ECG were 0.1204 and 0.0108, respectively, and for the LF/HF ratio, the *p*-values were also 0.0403 and 0.0627 for radar and ECG, respectively. These results reveal that certain radar-derived HRV indices, particularly SDNN and mean RR, which demonstrate a high degree of similarity with ECG HRV indices based on Figure 7, exhibit statistically significant differences between the high-mental distress and control groups, especially when considering the conventional *p*-value threshold of 0.05.

### 3.2. Receiver Operating Characteristic (ROC) Analysis

In our study, we generated receiver operating characteristic (ROC) curves based on mean RR and SDNN values obtained using ECG and radar. ROC curves are typically employed to assess the performance of binary models. They show the tradeoff between sensitivity (true positive rate) and 1-specificity (false positive rate). ROC curves allow visual inspection of how the model performs at various thresholds. In our study case, we utilized HRV indices (mean RR, SDNN) to distinguish between a high-mental distress group and a control group. The ROC curves are represented in Figure 9.

The ROC curves demonstrate the effectiveness of radar-based HRV measurements as a distinguishing tool for the high-mental distress group by showing comparable performance to ECG across key metrics. In each graph, the AUC (area under the curve) represents the area under the ROC curve, providing a quantitative measure of the overall performance of the model in distinguishing between the high-mental distress group and the control group. AUC values range from 0 to 1, with higher values typically signifying superior model performance. The AUC values for radar-based mean RR and SDNN measurements were 0.730 and 0.718, respectively, indicating “acceptable model” performance in distinguishing high mental distress, and these values are similar to ECG’s, which were 0.731 for mean RR and 0.753 for SDNN.

The values represented by the red asterisk on the graph correspond to the “optimal threshold value (specificity, sensitivity)” for each method. The optimal threshold was selected using “top left” methods in the pROC package in R studio, which aims to find the threshold that is closest to the top-left corner of the ROC curve. This corner represents the point where sensitivity is maximized, and 1-specificity is minimized simultaneously. The threshold for radar mean RR was 781.964, which is similar to the ECG threshold of 781.150 and demonstrated moderate levels of specificity (0.700) and sensitivity (0.667). The analysis of the SDNN values from ECG and radar measurements revealed distinctive strengths in mental distress recognition, while the mean RR from ECG and radar showed very similar trends. For radar SDNN, the threshold was 34.456, showing significant differences from the ECG threshold of 26.714 and poor specificity (0.511) according to top-left model. Selecting a different threshold could improve specificity. However, the radar system was not superior in specificity compared to ECG; it was, in fact, superior in sensitivity. ECG, with its SDNN metric, demonstrated proficiency in identifying true negatives, indicating individuals without mental distress. In essence, ECG was particularly effective in recognizing individuals who genuinely did not exhibit signs of mental distress. Conversely, when considering the SDNN values from radar measurements, it excelled amazingly in recognizing true positives, and accurately identifying individuals with high levels of mental distress. Surprisingly, radar was notably adept at detecting individuals who indeed were experiencing heightened mental distress.

In summary, both radar HRVs presented valid decision-making, appropriate capabilities with AUC > 0.7. ECG stood out for recognizing those without mental distress (true negatives), while radar excelled in identifying individuals experiencing significant mental distress (true positives). Notably, the radar exhibited a distinct pattern that prooved advantageous for accurate identification in true positive cases. The statistical results from the experiments, coupled with the convenience of measurement, confirm radar’s emergence as a more useful tool for detecting individuals with elevated levels of mental distress.

## 4. Discussion

The experiment was designed to estimate mental distress using IR-UWB radar in a software development company where employees could be at risk of high levels of distress. To validate HRV measurements using IR-UWB radar, a total of 102 experiments were conducted, excluding outliers. HRV indices were obtained from both the IR-UWB radar and ECG sensor, and their correlations were subsequently analyzed. Furthermore, a group analysis was performed to examine significant differences among the categorized groups (high-mental distress group and control group) based on the VAS scores. In this section, we examine and interpret the findings, highlighting their implications and acknowledging the limitations of our research work.

The HRV indices obtained from both ECG and IR-UWB radar exhibited a strong correlation, indicating a high level of agreement between the two measurement methods. Specifically, the mean RR values showed remarkable similarity, with a Pearson’s correlation coefficient of 0.9997. Additionally, the SDNN, RMSSD, and LF/HF ratio indices displayed significant correlations [43], as shown in Figure 7. The Bland–Altman plot indicates that the mean RR interval difference was 0.05 ms, with narrow limits of agreement (+5.2 and −5.1 ms), suggesting minimal average deviation and relatively consistent measurements. For SDNN, the mean difference was 3.0 ms, with limits of agreement ranging from +15 to −9.4 ms, indicating a moderate level of variability. In contrast, RMSSD showed a mean difference of 11.0 ms, with wider limits of agreement (+53 to −32 ms), highlighting significant variability and error, especially as RMSSD values increased. The LF/HF ratio had an offset of −0.14, with limits of agreement between +0.33 and −0.61 SD, reflecting a moderate spread. Overall, while the mean RR interval and SDNN were relatively consistent, RMSSD demonstrated more variability [44,45,46], and the LF/HF ratio showed a moderate level of deviation. 

Compared to state-of-the-art research, our study exhibits several differences. Specifically, the accuracy of our study’s results appears to be lower than that of previous studies [47,48,49]. However, it is important to note that the previous studies predominantly utilized continuous wave (CW) or frequency-modulated continuous wave (FMCW) radar systems for HRV calculation, which are significantly more expensive than the radar system used in our study. The high cost associated with CW radar presents a significant barrier to the commercialization of radar-based HRV monitoring. Additionally, the studies referenced have a limited number of evaluations, with a maximum of 30 evaluations, making it challenging to directly compare the accuracy levels between our study and others.

Notably, the radar measurements in this study were conducted while the participants were in a seated position, which differs from previous research [50]. This highlights the feasibility of using radar for HRV measurement in a variety of settings and conditions. The correlation analysis revealed sufficient accuracy of the radar-based HRV measurement, leading us to conclude that radar-based HRV measurements can effectively estimate mental distress. 

To assess mental distress through radar-based HRV, we investigated the relationships among HRV indices. We performed a correlation analysis between radar-derived HRV indices and mental distress, revealing notable trends. Several HRV indices measured via radar demonstrated negative correlations with VAS scores: mean RR had a correlation coefficient of −0.2644, SDNN was −0.2257, and RMSSD was −0.1334, based on 102 evaluation samples. In contrast, spectral measures exhibited a positive correlation, with the LF/HF ratio showing a coefficient of 0.1392. These results are consistent with previous research, which has shown that reduced HRV—indicated by lower values in mean RR, SDNN, and RMSSD—is often linked to psychiatric conditions such as depression and anxiety [11,12]. 

Table 2 presents the average values of HRV indices for each of the four groups. For mean RR and SDNN, both radar and ECG measurement showed relatively stable values across the three control groups. However, the mean RR and SDNN showed a notable decrease compared to the control groups, with both radar (mean RR: 727.87 ms, SDNN: 26.61) and ECG (mean RR: 727.27 ms, SDNN: 22.28). These results reinforce the radar’s reliability in detecting significant HRV changes associated with mental distress. Although RMSSD and LF/HF ratio values derived from radar differed significantly from those measured by ECG, both devices showed a similar reduction in RMSSD for the high-mental distress group. While radar provided higher RMSSD values than ECG, both methods indicated a clear decline in the high-mental distress group. The LF/HF ratio showed an increase in the high-mental distress group compared to control groups, unlike other indices. This trend is consistent with the expected trend that mental distress is often associated with increased sympathetic nervous system activity. We conducted an ANOVA analysis on each HRV index, extending beyond simple average comparisons. As depicted in the box plot in Figure 8, the analysis revealed significant group differences for several indices. Both mean RR and SDNN showed *p*-values below 0.05 for both radar and ECG measurements, indicating distinct distribution patterns between the high-mental distress and control groups. However, the radar-derived RMSSD, with a higher p-value of 0.1204, lacked the precision of its ECG counterpart, making it difficult to differentiate the high-mental distress group using radar-based RMSSD. While the radar-derived LF/HF ratio had a lower *p*-value than ECG, the relatively weak correlation between the two tools suggests that using radar-based LF/HF to assess mental distress may not be reliable. Nevertheless, the findings suggest the potential to distinguish individuals with severe mental distress solely through mean RR and SDNN measured using IR-UWB radar.

Figure 9’s ROC analysis of SDNN values from both ECG and radar measurements highlights their distinct roles in recognizing mental distress. ECG’s proficiency in identifying true negatives aligns with its established role in cardiovascular health monitoring, making it effective for recognizing individuals without mental distress. Conversely, radar, particularly in SDNN values, excels in identifying true positives, showcasing its adeptness in accurately detecting heightened mental distress. The observed distinct pattern in radar further enhances its utility, positioning radar as a more useful tool for identifying individuals experiencing significant mental distress. In summary, the combined strengths of both ECG and radar contribute to valid decision-making capabilities, with radar emerging as particularly valuable in scenarios demanding heightened sensitivity to mental distress.

However, this study has several limitations. First, a notable proportion of outliers, accounting for up to 15%, occurred during the 120 radar-based HRV measurements, resulting in 18 unsuccessful HRV measurements. Therefore, future research should explore more robust HRV measurement methods to minimize outliers. This study also involved repeated measurements of a single subject, with some measurements removed due to outliers. This may have introduced distortion because of the imbalance in the number of trials for each person. To address this issue, we conducted further analysis, and there were no significant deviations from the original trends, indicating that the results were not substantially affected. As shown in Appendix A, the trends in the averaged results, which account for errors from repeated measurements, align closely with the original findings, confirming that the results were not significantly impacted. Additionally, the VAS scores themselves were independently measured. Therefore, each sample maintains a reasonable level of independence, making it valid to treat them as independent measurements. To verify this, we assessed intra-individual differences but found no significant variation from the original analysis of VAS scores and HRV indices. As a result, the mental distress evaluation using VAS can still be considered independent, as each VAS measurement was conducted separately for each evaluation. The potential errors introduced by repeated measurements did not significantly influence the overall trends or outcomes in the study.

Moreover, it is essential to recognize the limitations inherent to radar measurements. Not all electrical changes measurable through ECG manifest as physical movements of the heart. Consequently, radar, which primarily captures information related to physical movement such as displacement and speed, has inherent structural limitations. To achieve a time resolution of 4 ms in this work, HRV was recorded at a frame rate of 250 FPS. However, because of the high FPS, the operating properties of the radar deteriorated, resulting in signal-to-noise ratio (SNR) degradation and increased phase noise. To achieve practical applications such as a one-chip solution, real-time signal processing to address these performance limitations would have to be implemented. However, the high FPS of 250 would make it difficult to integrate signal processing into a real-time operating system. The experimental design of the study also has limitations. Due to the directional constraints of the radar antenna, our study focused on observing the subject’s upper body, limiting the availability of sedentary movement information. Furthermore, considering the radar’s sensitivity to movement, participants were instructed to minimize unnecessary movements during the experiments to ensure accurate HRV measurement. Consequently, it is possible that such instructions or movement restrictions might have influenced changes in HRV and the level of mental distress, potentially impacting the study’s results and analysis. In our study, evaluations were classified based on the VAS score, with a score of 7.5 or higher designated as belonging to the high mental distress group. However, within the 102 evaluations, only 12 evaluations were classified into this high-mental distress category. This substantial imbalance in the distribution of high-mental distress data could potentially impact the precision and reliability of our study findings.

Despite the limitations, this study demonstrates the feasibility of measuring HRV using radar and assessing mental distress using HRV. The validity of HRV measured by radar has been verified, and it correlates with the level of mental distress. Moreover, the study has validated the accuracy of HRV measurements in a sitting position with an improved fast-time localization algorithm. Non-contact HRV measurement using radar overcomes the constraints of traditional contact-based sensors, enabling routine and long-term HRV monitoring. Using these measurement properties, radar can be utilized as an early screening tool for identifying groups with high mental distress. In a nutshell, implementing radar-based stress monitoring contributes to the overall improvement of occupational health by addressing stress factors in real work environments, which is particularly beneficial for highly sedentary professions like software development. Early detection of high-stress cases enables proactive intervention measures, promoting employee well-being and mitigating potential long-term health impacts. Radar-based stress measurement provides real-time insights into individuals’ stress levels, allowing for timely intervention and support.

## 5. Conclusions

We conducted a feasibility study utilizing IR-UWB radar-based HRV parameters to predict high-level mental distress among employees working in a software startup company. This investigation involved measuring HRV parameters using an IR-UWB radar sensor and evaluating 120 evaluations for assessing high-level mental distress. After excluding outlier events, 102 evaluations were subjected to analysis, revealing a strong positive correlation between HRV indices obtained from the IR-UWB radar and ECG sensor. Furthermore, our study identified a negative correlation between mental distress levels and HRV, with participants experiencing high mental distress showing significantly lower mean RR intervals and SDNN values compared to the control group. Remarkably, radar-based analysis demonstrated substantial discriminatory capability in identifying individuals with genuine high mental distress, as supported by our robust ROC analysis. 

However, no participants in the study had been diagnosed with depression or anxiety disorders, which may have limited the statistical power of the study. Furthermore, the imbalance in the number of evaluations between the high-mental distress group and other groups, along with the repeated measurements of each subject, could have influenced the results. To address this limitation, future research should aim to diversify the participant sample by including individuals from a diagnosed group. Additionally, efforts should be made to address the occurrence of outliers in HRV measurements.

Nevertheless, this paper demonstrates the sufficient validity of HRV measurements obtained with conventional IR-UWB radar and highlights the potential of HRV parameters measured by non-invasive approaches as a valuable early screening tool for identifying individuals with high levels of mental distress. This approach is particularly valuable for individuals needing daily stress monitoring where conventional HRV measurement methods are inconvenient or impractical. Furthermore, radar-based tools can also offer clinicians supplementary insights, enhancing decision-making by providing reliable data on autonomic nervous system activity, particularly in cases where traditional data is limited. This technology enables personalized stress management and mental health interventions for employees, helping to reduce burnout and improve mental health outcomes.

## Figures and Tables

**Figure 1 sensors-24-06210-f001:**
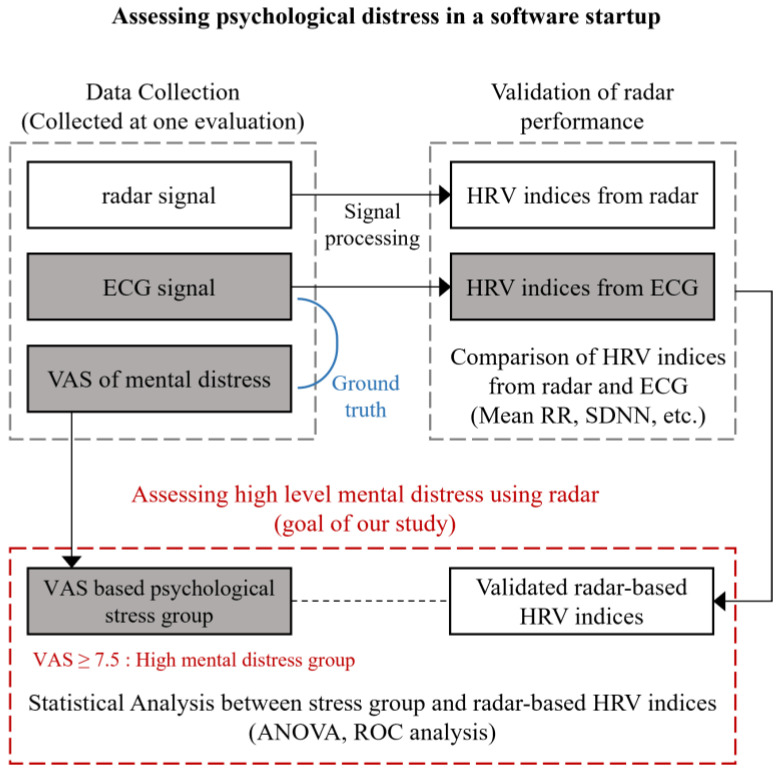
Structure of the study.

**Figure 2 sensors-24-06210-f002:**
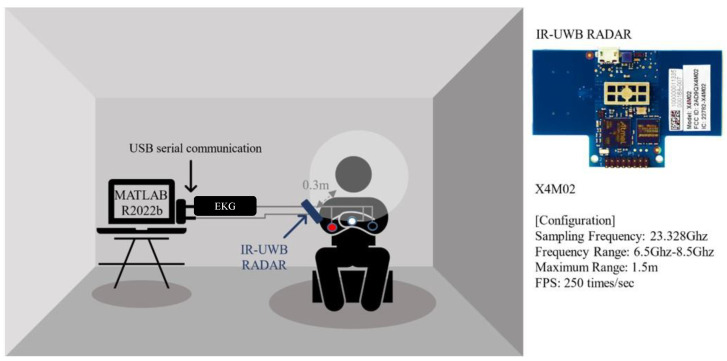
Experimental environment and detailed configuration of radar sensor.

**Figure 3 sensors-24-06210-f003:**
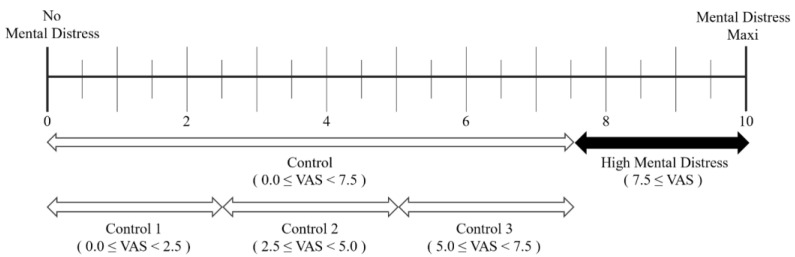
The VAS used in our experiment to evaluate mental distress.

**Figure 4 sensors-24-06210-f004:**
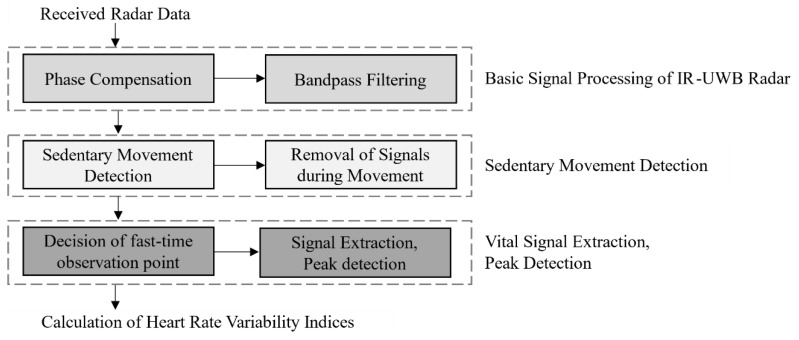
Signal processing diagram of the IR-UWB radar.

**Figure 5 sensors-24-06210-f005:**
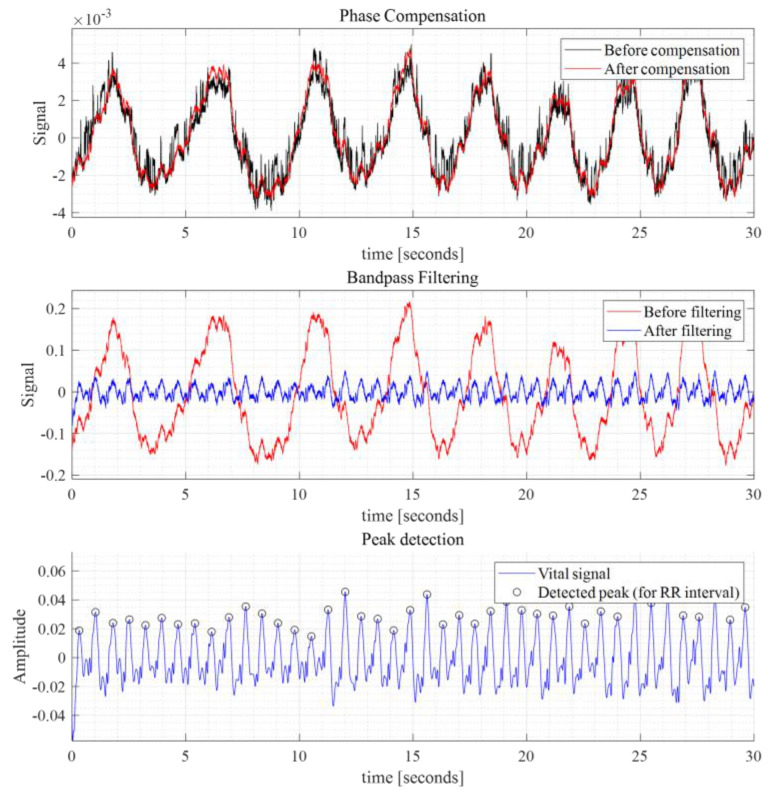
Intermediate signals for calculating HRV indices.

**Figure 6 sensors-24-06210-f006:**
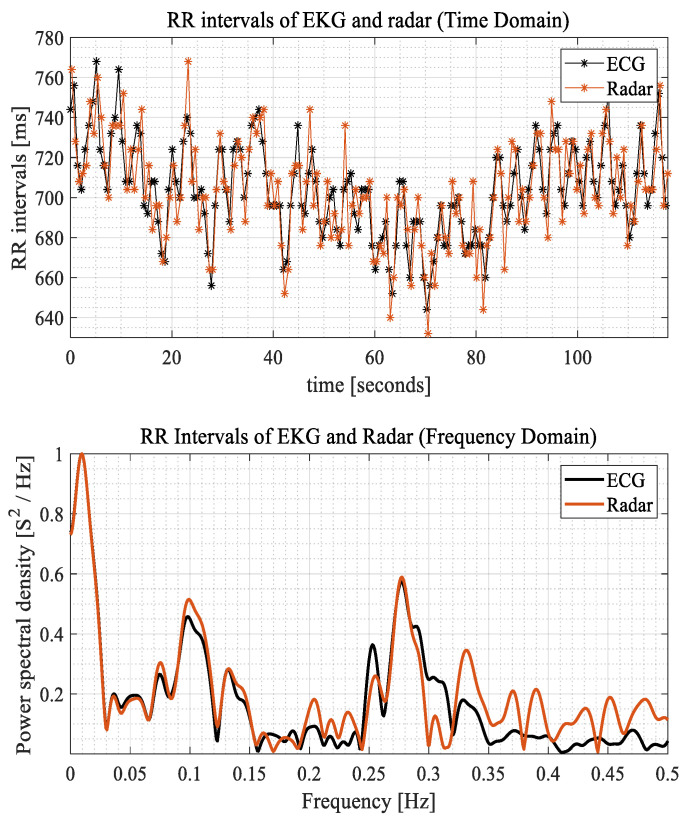
A series of peak (RR) intervals obtained from ECG and radar sensor.

**Figure 7 sensors-24-06210-f007:**
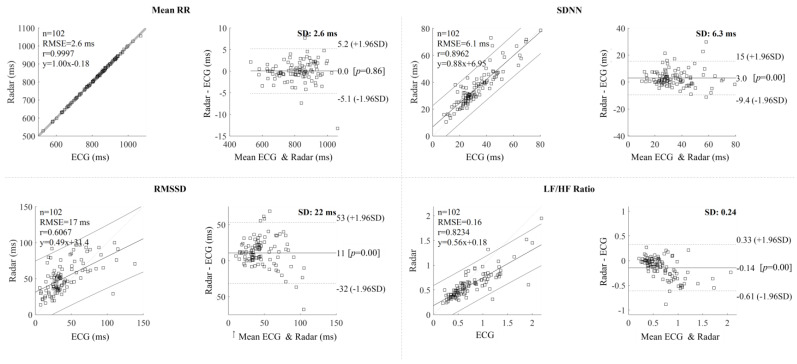
Scatter plots and Bland–Altman plots of the HRV parameters.

**Figure 8 sensors-24-06210-f008:**
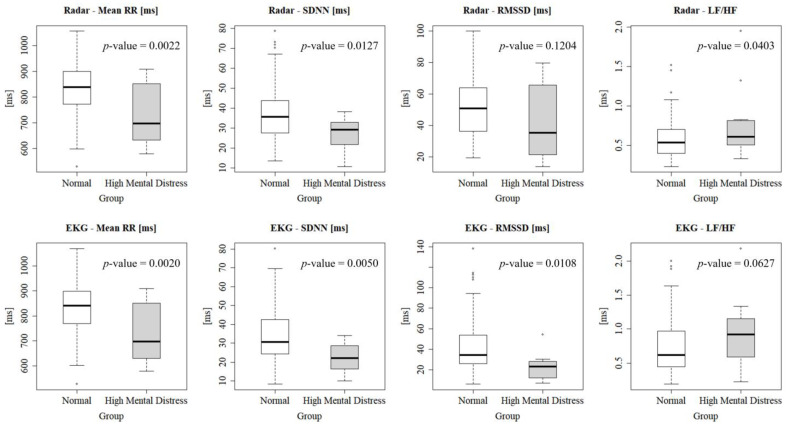
Box plots showing the distribution of HRV by group.

**Figure 9 sensors-24-06210-f009:**
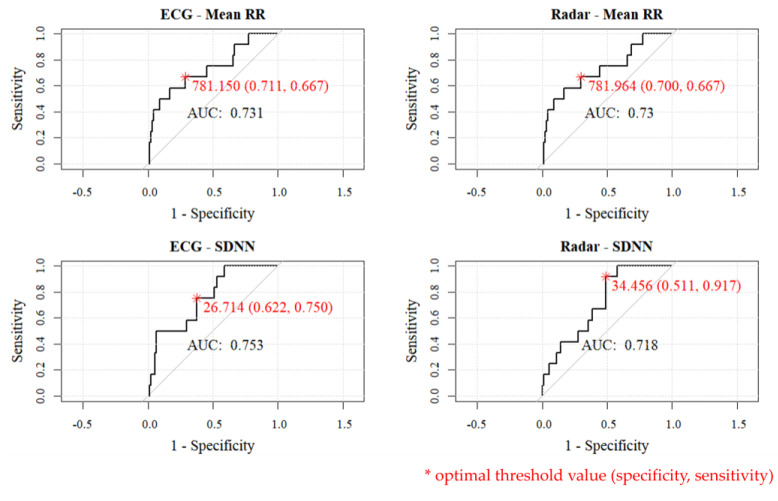
ROC curves of the HRV indices (left: ECG, right: radar).

**Table 1 sensors-24-06210-t001:** Demographic information.

Demographic Information	Mean	Median	Min	Max	Std. Dev
Age	30.9	30.0	23	44	4.565
Height	1.72	1.73	1.59	1.83	0.059
Weight	72.8	73.8	53.6	87.7	10.41
BMI	24.3	23.3	21.2	29.3	2.773
Gender	Male: 12, Female: 3				
Job Roles	Software Engineer: 11, Production Manager: 2, 3D Designer: 1, Office Administrator: 1				

**Table 2 sensors-24-06210-t002:** Average value of HRV indices for each group measured by both ECG and radar.

Group	IR-UWB Radar				ECG			
	Mean RR [ms]	SDNN [ms]	RMSSD [ms]	LF/HF Ratio	Mean RR [ms]	SDNN [ms]	RMSSD [ms]	LF/HF Ratio
Control 1	840.61	37.71	50.35	0.58	840.76	34.99	39.28	0.71
Control 2	831.81	37.86	62.27	0.56	832.15	36.31	51.10	0.72
Control 3	822.08	36.50	49.86	0.59	821.93	33.10	41.59	0.73
Control (all) ^1^	828.56	37.08	52.73	0.58	828.59	34.25	43.16	0.72
High mental distress	727.87	26.61	42.50	0.76	727.27	22.28	22.76	0.95

^1^ Control (all) = Sum of all control groups.

## Data Availability

The data presented in this study are available on request from the corresponding author.

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
