# Peer review of "Feasibility of Early Assessment for Psychological Distress: HRV-Based Evaluation Using IR-UWB Radar"

_sensors, 2024, doi:10.3390/s24196210_

Round 1

Reviewer 1 Report

Comments and Suggestions for Authors

General Overview:

 The study presents a novel approach to assess mental distress using radar-based HRV (Heart Rate Variability) measurements. While the concept is well-defined, several sections require refinement for clarity, methodological transparency, and grammatical accuracy.

 Abstract (Lines 11-26):

    Line 15: The phrase "aimed to early assess" should be restructured for clarity. Consider changing it to "aimed at assessing early high levels of mental distress."

    Line 23: The term "discriminatory ability" can be misinterpreted. Consider rephrasing it to "ability to differentiate."

    Line 25: The term "adjuvant diagnostic information" is vague. You could replace it with "complementary diagnostic insights."

Introduction (Lines 30-133):

    Line 36: The sentence "has been utilized" should be "have been utilized" when referring to "methods."

    Line 43: The phrase "in which anxiety disorders are associated" should be simplified. Consider "where anxiety disorders are linked to decreased HRV."

    Line 104: "Pre-clinical validation" might not be appropriate as the study involves human subjects. Authors could use "preliminary validation" instead.

    Lines 107-109: The repetition of "radar-based" can be avoided by merging sentences: "This novel approach assesses high levels of mental distress through HRV measurement using radar technology and VAS scores."

Materials and Methods (Lines 134-318):

    Line 153: "The demographic information table" is redundant. Consider writing "Demographic information is provided in Table 1."

    Line 165: "120 experiment were conducted" should be corrected to "120 experiments were conducted."

    Line 173: Clarify what "radar transceiver pair" refers to, including more specifics about the nature and configuration of the transceivers.

    Line 174: The technical setup is explained well but would benefit from more detail about radar configuration, error margins, and noise sensitivity.

    Line 237-242: The description of phase compensation could be clearer, particularly regarding how phase noise is managed differently in this study compared to standard setups. Expanding on the noise reduction techniques would provide better context.

    Line 272: The paper should provide explicit frequency boundaries for the low-frequency (LF) and high-frequency (HF) bands used in the calculation of the LF/HF ratio. For clarity, also specify further details regarding the frequency domain analysis performed, including the interpolation method used, handling of transients, and other relevant aspects.

Results (Lines 320-396):

    Line 331: "Bland Altman plots" should be "Bland-Altman plots" (with a hyphen).

    Lines 366-369: The ROC curve descriptions are sufficient, but further explanation could help non-experts. How do these curves demonstrate the effectiveness of HRV as a diagnostic tool for mental distress?

Discussion (Lines 397-483):

    Line 418: The phrase "radar can be used to estimate mental distress" is vague. Instead, say "radar-based HRV measurements can effectively estimate mental distress."

    Line 421: There’s a minor contradiction when saying HRV indices correlate with mental distress without fully explaining how radar differs from ECG in precision. The differences between radar and ECG in terms of HRV measurement precision should be more thoroughly explored.

Conclusion (Lines 484-510):

    Line 497: The phrase "weakened the statistical distinctions observed" is unclear. Consider changing it to "may have limited the statistical power of the study."

    Line 503: The sentence "valuable early screening tool for estimating the high level of mental distress" could be clearer. Consider using "valuable early screening tool for identifying individuals with high levels of mental distress."

Comments on the Quality of English Language

The quality of the English language has been addressed within the 'Comments and Suggestions for Authors' section.

Reviewer 2 Report

Comments and Suggestions for Authors

Although a interesting promising technique, the text should be improved. There are many negligences in the text (like 'ECG is a medical test' in line 49) and the introduction is too long. The method does not provide information of the ECG device, except it's name (PSL-iECG2). The time resolution, peak detection method should be provided. For both methods, artefact correction procedures must be described (and should be the same for both methods for a fair comparison). Also information is missing about the synchronisation of ECG and radar signal and the used method for spectral analysis.

RMSSD has poor correlation and shows large differences in the B&A plot. This measure is most sensitive to beat-beat errors, but is hardly evaluated. Authors should relate these results to the differences in ECG and PPG and/or finger pressure methods to measure HRV. Since they in fact measure carotid pressure pulse, which causes the distance to change, this comparison will be in the advantage of the radar method.

Spectral measures and RMSSD are missing in the 'HRV indices and VAS score' section.

Repeated measurements of one subject are treated as independent, which is not correct. How to deal with that, is decribed by Bland and Altman.

I was surprised to read the text 'An increased heart rate is often associated with a decrease in the mean RR interval' (line 421). It's strange that four authors agreed to this faulty statement.

Reviewer 3 Report

Comments and Suggestions for Authors

The paper presents the feasibility study for the contactless assessment of mental distress. The study was performed on 15 subjects employed in the software development company. The experimental protocol consists of 5 min recordings of electrocardiogram (ECG, as a ground truth signal) and cardiac movement using the IR-UWB radar, twice a day (at the beginning and the end of the working day). This recording was repeated in four days for each subject, so the total number of recordings was 120. Subjective assessment using the Visual Assessment Scale (VAS) was performed for each physiological recording. The authors used 102 of 120 recordings for further analysis that included comparison of radar vs ECG extracted HRV indices (Mean RR, SDNN, RMSSD, LF/HF). The alignment between radar and ECG HRV indices was presented using Bland Altman analysis. Also, boxplots and ROC analysis for Mean RR and SDNN were presented for radar and ECG recordings. The methodology and discussion are quite detailed, but some aspects should be further clarified:

11. What was the window for averaging RR intervals?

22. Section 2.4 Radar Signal Processing and Algorithm is an important part of the methodology and in this section, all steps should be more clarified and illustrated step by step on a radar signal example.

33. Sedentary movements were detected, and the signals recorded during sedentary movements were not used in further analysis. What was the percentage of the excluded signal? How did you calculate HRV parameters on the signal with excluded parts? Did you perform interpolation?

44. What is the accuracy of HRV indices extracted from radar signal? Can you compare it with the state-of-the-art?

55.  Table 2 shows a large disbalance of high mental distress data vs control data (12 recordings vs 90 recordings). This fact should be discussed and included in the study limitation.

Round 2

Reviewer 2 Report

Comments and Suggestions for Authors

The revised version has greatly improved and is acceptable for publication. I forgot to mention that there is one subject missing in the list of Job roles in table 1. Beside this small detail, I'm pleased with the answers.

Author Response

Thank you for your feedback and for accepting the revised version for publication. We greatly appreciate your kind words.

We have promptly updated Table 1 to include the missing subject from the list of Job roles as you mentioned. Specifically, we had missed one male who is a software engineer, so the count of software engineer has been updated from 10 to 11. These changes can be found in Table 1 at line 149.

Reviewer 3 Report

Comments and Suggestions for Authors

Authors have properly addressed all my comments. The paper is significantly improved.

Author Response

We are glad to hear that the revisions have properly addressed all your comments! Your insights were invaluable in refining the manuscript, and we truly appreciate your time and effort.